# Mitochondrial respiration atlas reveals differential changes in mitochondrial function across sex and age

Dylan C Sarver[1,2], Muzna Saqib[1,2], Fangluo Chen[1,2], G William Wong[1,2]*

[1]Department of Physiology, Johns Hopkins University School of Medicine, Baltimore, United States; [2]Center for Metabolism and Obesity Research, Johns Hopkins University School of Medicine, Baltimore, United States

## eLife Assessment

This **important** study provides a comprehensive assessment of mitochondrial function across age and sex in mice. The strength of evidence supporting this resource is **compelling**, given the exhaustive number of tissues profiled and in-depth analyses performed.

*For correspondence:
gwwong@jhmi.edu

Competing interest: The authors declare that no competing interests exist.

**Abstract** Organ function declines with age, and large-scale transcriptomic analyses have highlighted differential aging trajectories across tissues. The mechanism underlying shared and organ-selective functional changes across the lifespan, however, still remains poorly understood. Given the central role of mitochondria in powering cellular processes needed to maintain tissue health, we therefore undertook a systematic assessment of respiratory activity across 33 different tissues in young (2.5 months) and old (20 months) mice of both sexes. Our high-resolution mitochondrial respiration atlas reveals: (1) within any group of mice, mitochondrial activity varies widely across tissues, with the highest values consistently seen in heart, brown fat, and kidney; (2) biological sex is a significant but minor contributor to mitochondrial respiration, and its contributions are tissue-specific, with major differences seen in the pancreas, stomach, and white adipose tissue; (3) age is a dominant factor affecting mitochondrial activity, especially across most brain regions, different fat depots, skeletal muscle groups, eyes, and different regions of the gastrointestinal tract; (4) age effects can be sex- and tissue-specific, with some of the largest effects seen in pancreas, heart, adipose tissue, and skeletal muscle; and (5) while aging alters the functional trajectories of mitochondria in a majority of tissues, some are remarkably resilient to age-induced changes. Altogether, our data provide the most comprehensive compendium of mitochondrial respiration and illuminate functional signatures of aging across diverse tissues and organ systems.

## Introduction

Aging is a complex biological phenomenon that results in functional decline across tissues and organ systems (*López-Otín et al., 2013*). Age is one of the most significant contributors to disease risk (*Kennedy et al., 2014*), and the aging process is influenced by a variety of genetic and environmental factors at the tissue and organismal level (*Tian et al., 2023*; *Nie et al., 2022*; *Kuo et al., 2022*). The application of omics technologies in recent years has led to unprecedented insights into the complex biology of aging (*Rutledge et al., 2022*). Bulk and single-cell transcriptomic analyses of mouse tissues across the lifespan have shown that aging tempo and trajectory, as indicated by tissue transcriptomic signatures, varies widely across tissues (*Schaum et al., 2020*; *Zhang et al., 2021*; *Almanzar et al., 2020*). Remarkably, many of the transcriptomic signatures of aging across tissues can be significantly

reversed by caloric restriction or rejuvenated by the transfusion of young blood (*Pálovics et al., 2022*; *Ma et al., 2020*). Large-scale proteomic analyses of human plasma also reveal distinct waves of changes across the lifespan, which are associated with distinct biological pathways and affect age-related phenotypic traits and diseases (*Lehallier et al., 2019*; *Oh et al., 2023*).

Despite enormous progress, there is no consensus regarding the mechanisms underlying aging at the cellular or tissue level. Although many non-mutually exclusive hypotheses have been put forth to explain the root cause of aging - oxidative damage, genomic instability, epigenetic changes, loss of proteostasis, mitochondrial dysfunction, DNA damage, telomere shortening, cellular senescence, stem cell exhaustion - it remains a major challenge to distinguish between the driver and passenger mechanisms of aging (*de Magalhães, 2024*). Nevertheless, efforts to understand the proximal and ultimate cause of aging will facilitate development of therapeutics aimed to improve healthy aging (*Kaeberlein et al., 2015*; *Campisi et al., 2019*).

In the present study, we focused on aging from a mitochondrial perspective, as this organelle is known to play an important role in the aging process (*Jang et al., 2018*; *Lima et al., 2022*; *Lanza and Nair, 2010*; *Sun et al., 2016*). Mitochondria supply the bulk of the energy needed to maintain tissue health and repair tissue damage, and their function tends to decline with age. Over time, damage accumulates in mitochondrial DNA, proteins, and lipids, which compromises their functional integrity and leads to dysregulated metabolism and increased oxidative stress. Recent transcriptomic analyses have highlighted major reductions in electron transport chain gene expression across the lifespan (*Schaum et al., 2020*), and these changes can be significantly reversed by the transfusion of young blood into older mice (*Pálovics et al., 2022*). Aging-associated reduction in mitochondrial OXPHOS gene expression appears to be conserved between human, mouse, fly, and worm (*Zahn et al., 2007*). Accordingly, mitochondrial dysfunction has been implicated in various age-related diseases, including neurodegenerative disorders, cardiovascular diseases, and metabolic syndromes (*Haas, 2019*). Boosting mitochondrial health has been shown to delay age-related decline in organ function (*Foote et al., 2018*; *Lima et al., 2023*; *Chiao et al., 2020*; *Nilsson and Tarnopolsky, 2019*).

Given the central role of mitochondria in tissue health, we aimed to address the extent and magnitude of aging-induced changes in mitochondrial function across tissues and organ systems. Although many studies have examined mitochondrial respiratory capacity in various tissues, the scale was limited in that only a very small number of tissues could be interrogated at the same time. This is largely due to the inherent low-throughput method of assessing respiration which requires freshly isolated mitochondria or cells from tissues (*Salabei et al., 2014*). Consequently, it was not previously feasible to have a comprehensive and systems-level analysis of mitochondrial function across many tissues and the lifespan.

This barrier, however, has been recently overcome. An innovative method by Acin-Perez and coworkers has made it possible to now assess mitochondrial function in previously frozen tissues (*Acin-Perez et al., 2020*). This new method circumvents the need to isolate mitochondria at the time of tissue harvest, allowing many tissues to be collected, frozen, and assayed at a later time. We adopted the new method in a standardized workflow to profile mitochondrial activity in 33 tissues from young and old mice of both sexes. The dataset consists of a total of 1320 tissue samples from 40 mice and 3960 high-resolution respirometry assays encompassing three technical replicates. Our study represents the largest and the most comprehensive tissue respirometry analysis to date. Our data provide an unprecedented view on the variations and changes in mitochondrial functional capacity across tissues, sex, and age, thus informing ongoing studies on the causes and consequences of aging.

## Results

### Pan-tissue mitochondrial respiration atlas overview, workflow, and analysis pipeline

To assay mitochondrial respiration and its maintenance across age and sex, we collected 33 tissues from young (2.5 months; ~18-year-equivalent in human) or old (20 months; ~65-year-equivalent in human) male or female mice (n=10 mice per age and sex). The tissues collected include different brain regions (hippocampus, cortex, cerebellum, and hypothalamus), different sections of the gastrointestinal (GI) tract (stomach, duodenum, ileum, jejunum, cecum, proximal colon, and distal colon), various

fat depots (gonadal, inguinal, mesenteric), different skeletal muscle groups (tongue, diaphragm, quadriceps complex, hamstrings, gastrocnemius, plantaris, and soleus), reproductive organs (testis and fallopian tubes), as well as liver, pancreas, heart atria and ventricles, spleen, kidney cortex and medulla, eyes, and skin (*Figure 1A*). Our aim was to provide a comprehensive systems-level view of mitochondrial respiration across tissues, sex, and age.

Our standardized workflow involved thawing, mincing, and homogenization of frozen tissue samples in buffer. Samples were centrifuged to pellet cell debris and the supernatant was collected for immediate protein and mitochondrial content quantification (via MitoTracker Deep Red [MTDR]). Sample respiration rates were then assayed using a Seahorse XFe96 Analyzer (*Figure 1B*). The basic respiration assay consisted of four sequential steps: first, baseline unstimulated measurements were obtained. Then, NADH was used to assess respiration via mitochondrial complex I (CI), or succinate was used to assess respiration via mitochondrial complex II (CII) in the presence of rotenone (Rot, a CI inhibitor). Following this, samples were exposed to rotenone and antimycin A (AA, a CIII inhibitor) to silence respiration. Then, TMPD (*N*,*N*,*N'*,*N'*-tetramethyl-*p*-phenylenediamine) in the presence of ascorbate was used to assess respiration through mitochondrial complex IV (CIV), via donation of electrons to cytochrome *c* (*Figure 1C*). Detailed information of the methodology can be found in the Materials and methods section, which closely follows the method first described by *Acin-Perez et al., 2020*. High-resolution respirometry data for each of the 33 tissues were used in all subsequent comparisons (*Figure 1—figure supplements 1–33*; *Figure 1—source data 1*; *Figure 1—source data 2*).

## Mitochondrial function across different organ systems in male and female mice

The first analysis made was within a group (male, female, young, or old) across all tissues. This allowed us to focus on shared and unique mitochondrial properties across different tissues within a single mouse system. Ranking young male or young female tissues by their respiration via CI (NADH-stimulated), CII (succinate-stimulated), or CIV (TMPD and ascorbate-stimulated) showed that both sexes have the greatest oxygen consumption in the heart atria and ventricles, brown adipose tissue (BAT), kidney cortex and medulla, and the lowest respiration in the colon (distal or proximal), plantaris muscle, jejunum, ileum, and mesenteric white adipose tissue (mesWAT) (*Figure 2A and B*).

Ranking old male tissues by their respiration via CI, CII, or CIV showed that the heart atria and ventricles, BAT, diaphragm muscle, kidney cortex, and medulla have the highest oxygen consumption, while the ileum, pancreas, skin, duodenum, stomach, distal colon, mesWAT, and inguinal white adipose tissue (iWAT) have the lowest (*Figure 2C*). Ranking all old female tissues by their respiration via CI, CII, or CIV showed that the heart atria and ventricles, BAT, kidney cortex, and diaphragm have the highest respiration, while the pancreas, duodenum, mesWAT, ileum, distal colon, eyes, and skin have the lowest (*Figure 2D*).

Our analyses revealed a wide range in mitochondrial respiratory capacity across nearly all tissues. We observed tissue-types of high similarity, such as the different regions of the brain (*Figure 2—figure supplement 1*) and kidney cortex and medulla (*Figure 2—figure supplement 2*), as well as those displaying degrees of functional heterogeneity, such as the heart atria and ventricles (*Figure 2—figure supplement 3*), different skeletal muscle groups (*Figure 2—figure supplement 4*), various white adipose tissue depots (*Figure 2—figure supplement 5*), and sections of the GI tract (*Figure 2—figure supplement 6*). Together, these data help illustrate both the heterogeneity and lack thereof in mitochondrial function across different tissues and organ systems. Remarkably, even when age and sex are controlled for, mitochondria within a single tissue-classification or organ system can have distinct respiration signatures.

## Sex differences in mitochondrial function in young mice

The first pan-tissue comparison of mitochondrial function across groups was made between young male and female mice. In this comparison, we focused on the effects of sex on mitochondrial respiration in young mice. A systems-level view of mitochondrial respiration via CI, CII, or CIV across 32 tissues (omitting reproductive tissues) showed that young male and female mice have similar distributions in oxygen consumption rate (OCR; *Figure 3A–C*, left panel). Individual tissue-level analysis of respiration through CI showed only two significant differences between young male and female

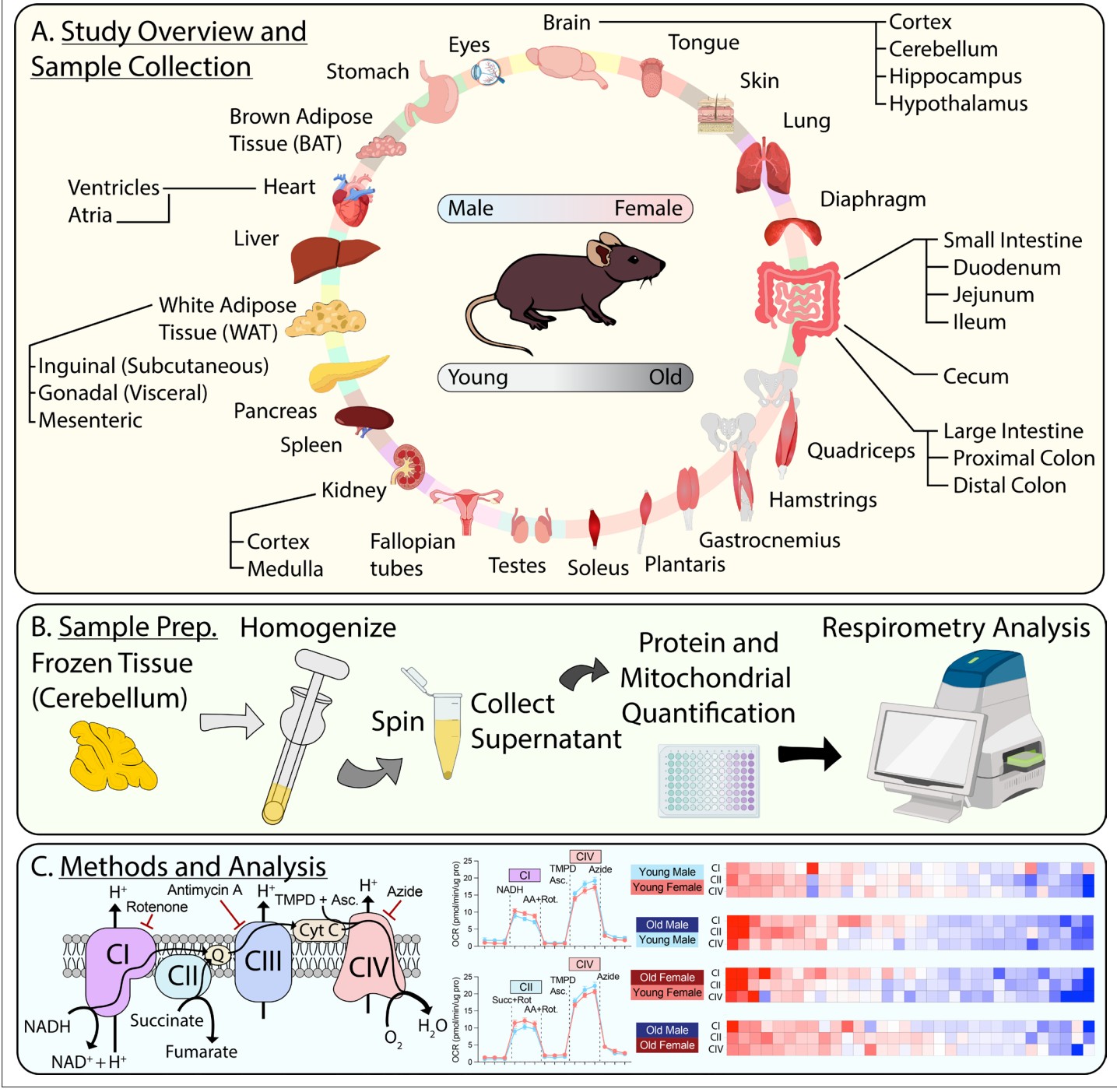

**Figure 1.** Pan-tissue mitochondrial respiration atlas overview, workflow, and analysis pipeline. (**A**) Study overview highlighting the 33 different tissues collected from four different groups of mice (n=10/group) for respirometry analysis and mitochondrial content quantification. The four groups of mice are young male, young female, old male, and old female. Young = 10-week-old; old = 80-week-old. (**B**) General schematic showing the preparation of samples for respirometry analysis. (**C**) General representation of the electron transport chain to illustrate the key components of the respirometry assay used to assess mitochondrial function, the associated data, and examples of subsequent data analysis. AA, antimycin A; Rot, rotenone; TMPD, N,N,N′,N′-tetramethyl-p-phenylenediamine; Asc, ascorbate; NADH, nicotinamide adenine dinucleotide.

The online version of this article includes the following source data and figure supplement(s) for figure 1:

**Source data 1.** Data for all tissues in old male and female mice.

**Source data 2.** Data for all tissues in young male and female mice.

**Figure supplement 1.** Mitochondrial respiration of brown adipose tissue (BAT) across sex and age.

*Figure 1 continued on next page*

*Figure 1 continued*

mice, in the quadriceps muscle complex and gonadal white adipose tissue (gWAT) (*Figure 3A*, right panel). Individual tissue comparisons of respiration through CII showed that young males have higher mitochondrial respiration in the kidney medulla, stomach, quadriceps muscle complex, hippocampus, and gastrocnemius muscle, and reduced mitochondrial activity in gWAT and ileum relative to young females (*Figure 3B*, right panel). Respiration at CIV showed that young males have higher mitochondrial activity in the BAT, heart ventricles, kidney medulla, hippocampus, stomach, quadriceps muscle complex, and gastrocnemius muscle and reduced mitochondrial activity in the gWAT and duodenum when compared to young females (*Figure 3C*, right panel). A relative and global view of the sex-specific differences across young male and female mice showed that males have higher mitochondrial respiration in the stomach, kidney, and skeletal muscle, whereas females have higher mitochondrial respiration in the GI tract and different fat depots (gWAT and mesWAT) (*Figure 3D*).

## Effects of age on male mitochondrial function

The second tissue-by-tissue comparison of mitochondrial function was made across age, where we focused on the effects of age on mitochondrial respiration in male mice. A systems-level view of respiration via CI, CII, or CIV across all 33 tissues showed that old and young male mice have similar

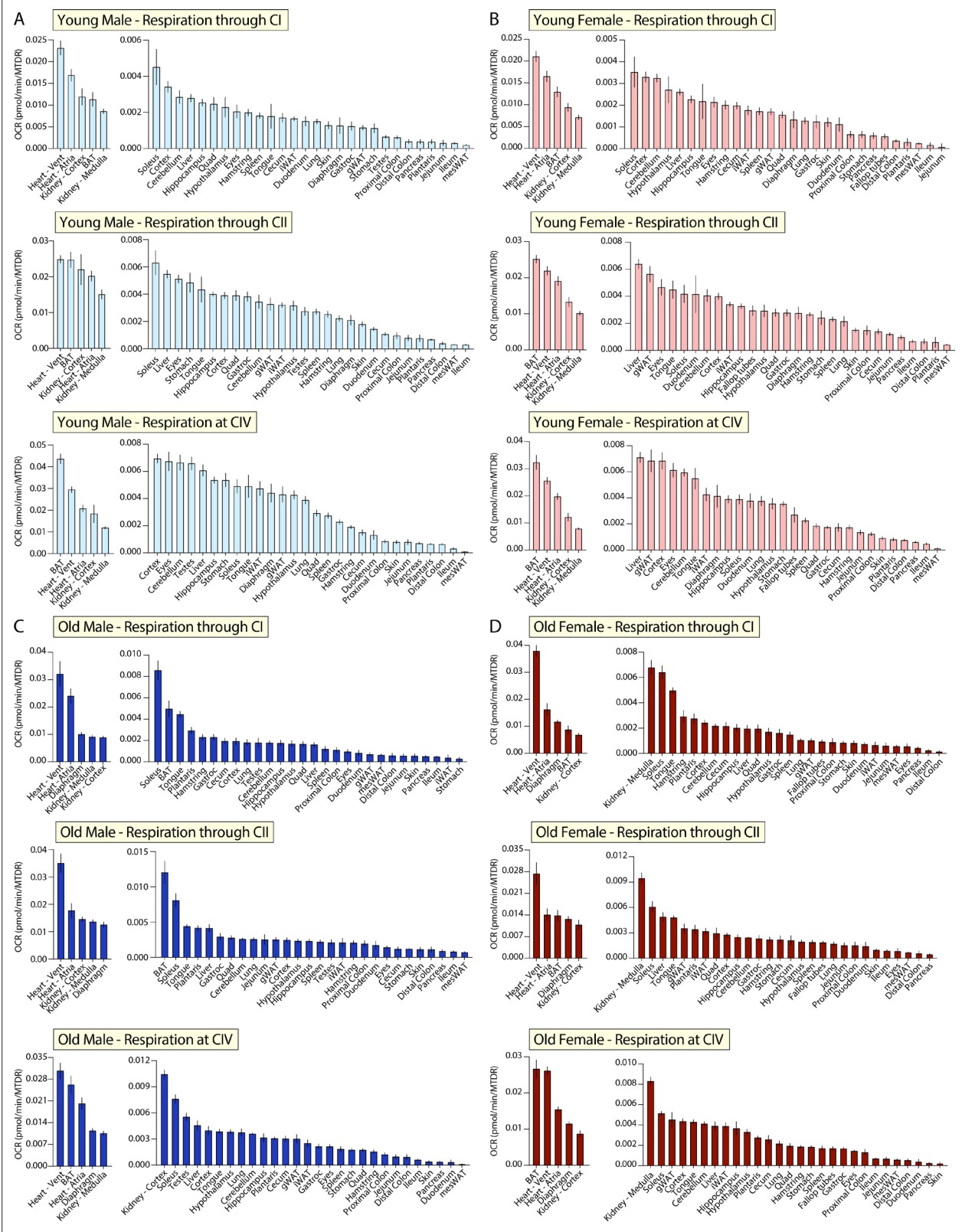

**Figure 2.** Tissue-by-tissue analysis of mitochondrial function in male and female mice. (**A**) Young male mitochondrial respiration through complex I (CI, NADH-stimulated), complex II (CII, succinate stimulated in the presence of rotenone, a CI inhibitor), and at complex IV (CIV, via TMPD+ascorbate). (**B**) Young female mitochondrial respiration through CI, CII, and at CIV. (**C**) Old male mitochondrial respiration through CI, CII, and at CIV. (**D**) Old female respiration through CI, CII, and at CIV, respectively. n=10 mice per tissue. All oxygen consumption rates (OCR) are normalized to mitochondrial content

*Figure 2 continued on next page*

*Figure 2 continued*

(based on MitoTracker Deep Red [MTDR]). All data is represented as the mean with standard error, and organized highest (left) to lowest (right). Young = 10 weeks; old = 80 weeks. BAT, brown adipose tissue; gWAT, gonadal white adipose tissue; iWAT, inguinal white adipose tissue; Quad, quadriceps muscles; Vent, ventricles; TMPD, *N,N,N′,N′*-tetramethyl-*p*-phenylenediamine.

The online version of this article includes the following figure supplement(s) for figure 2:

**Figure supplement 1.** Tissue-centric view of mitochondrial changes in the brain across sex and age.

**Figure supplement 2.** Tissue-centric view of mitochondrial changes in the kidneys across sex and age.

**Figure supplement 3.** Tissue-centric view of mitochondrial changes in the heart across sex and age.

**Figure supplement 4.** Tissue-centric view of mitochondrial changes in skeletal muscle across sex and age.

**Figure supplement 5.** Tissue-centric view of mitochondrial changes in the white adipose tissue across sex and age.

**Figure supplement 6.** Tissue-centric view of mitochondrial changes in the gastrointestinal (GI) tract across sex and age.

distributions of OCR (*Figure 4A–C*, left panel). However, an individual tissue view showed many differences in response to age. Old males showed a striking increase in mitochondrial activity via CI in the heart atria, skeletal muscles (diaphragm, soleus, tongue, plantaris, gastrocnemius), testis, mesWAT, and ileum, and reduced mitochondrial respiration in the brain cortex, cerebellum, hippocampus, liver, eyes, skin, iWAT, BAT, and stomach relative to young males (*Figure 4A*, right panel). Viewing respiration through CII also showed that old males have markedly higher mitochondrial activity in the heart ventricles, skeletal muscles (diaphragm and plantaris), jejunum, ileum, and mesWAT when compared to young males (*Figure 4B*, right panel). In contrast, old males had reduced mitochondrial activity via CII in the brain (cortex and hippocampus), BAT, stomach, quadriceps muscle complex, liver, and eyes when compared to young males. Respiration at CIV showed that old males have markedly elevated mitochondrial activity in the skeletal muscle (diaphragm, soleus, plantaris), ileum, and cecum, and reduced mitochondrial activity in the brain (cortex, cerebellum, hippocampus), BAT, liver, spleen, eyes, quadriceps muscle complex, pancreas, duodenum, stomach, and iWAT, relative to young males (*Figure 4C*, right panel). A relative and global view of the age-specific differences across male mice showed that old males have significantly elevated mitochondrial activity in the skeletal muscles, mesWAT, GI tract, and heart, with a concomitant reduction in mitochondrial activity in the stomach, iWAT, gWAT, BAT, eyes, and nearly all brain regions (*Figure 4D*). Together, these data indicate age has a strong effect in modulating mitochondrial respiration in male mice.

## Effects of age on female mitochondrial function

The third tissue-by-tissue comparison of mitochondrial function was made between old and young female mice. Systems-level analysis of respiration via CI, CII, or CIV across all 33 tissues showed a similar OCR distribution between old and young female mice (*Figure 5A–C*, left panel). Individual tissue comparisons, however, showed many significant differences in response to age. Respiration through CI showed that old females have elevated mitochondrial activity in the heart ventricles, skeletal muscles (diaphragm, soleus, tongue, and plantaris), and mesWAT, and a concomitant reduction in mitochondrial activity in the BAT, kidney cortex, brain (cortex and cerebellum), gWAT, iWAT, eyes, and distal colon when compared to young females (*Figure 5A*, right panel). Respiration through CII also showed that old females have increased mitochondrial respiration in skeletal muscles (diaphragm and plantaris), and mesWAT relative to young females (*Figure 5B*, right panel). In contrast, old females had reduced mitochondrial respiration via CII in the liver, brain (cortex and cerebellum), gWAT, fallopian tubes, eyes, skin, BAT, heart atria, and pancreas when compared to young females. Respiration at CIV likewise showed that old females have elevated mitochondrial activity in the skeletal muscle (diaphragm, plantaris, and soleus) and mesWAT relative to young females; concomitantly, old females had reduced mitochondrial activity in the heart atria, brain (cortex and cerebellum), liver, stomach, lung, eyes, jejunum, duodenum, pancreas, and skin when compared to young females (*Figure 5C*, right panel). A relative and global view of the age-specific differences across female mice showed that old females have elevated mitochondrial respiration in the skeletal muscles, mesenteric fat, and heart ventricles, and a concomitant reduction in mitochondrial activity in the eyes, skin, duodenum, gWAT, iWAT, BAT, and nearly all brain regions when compared to young females. Similar to males, these data also indicate age has a strong effect in modulating mitochondrial activity in female mice.

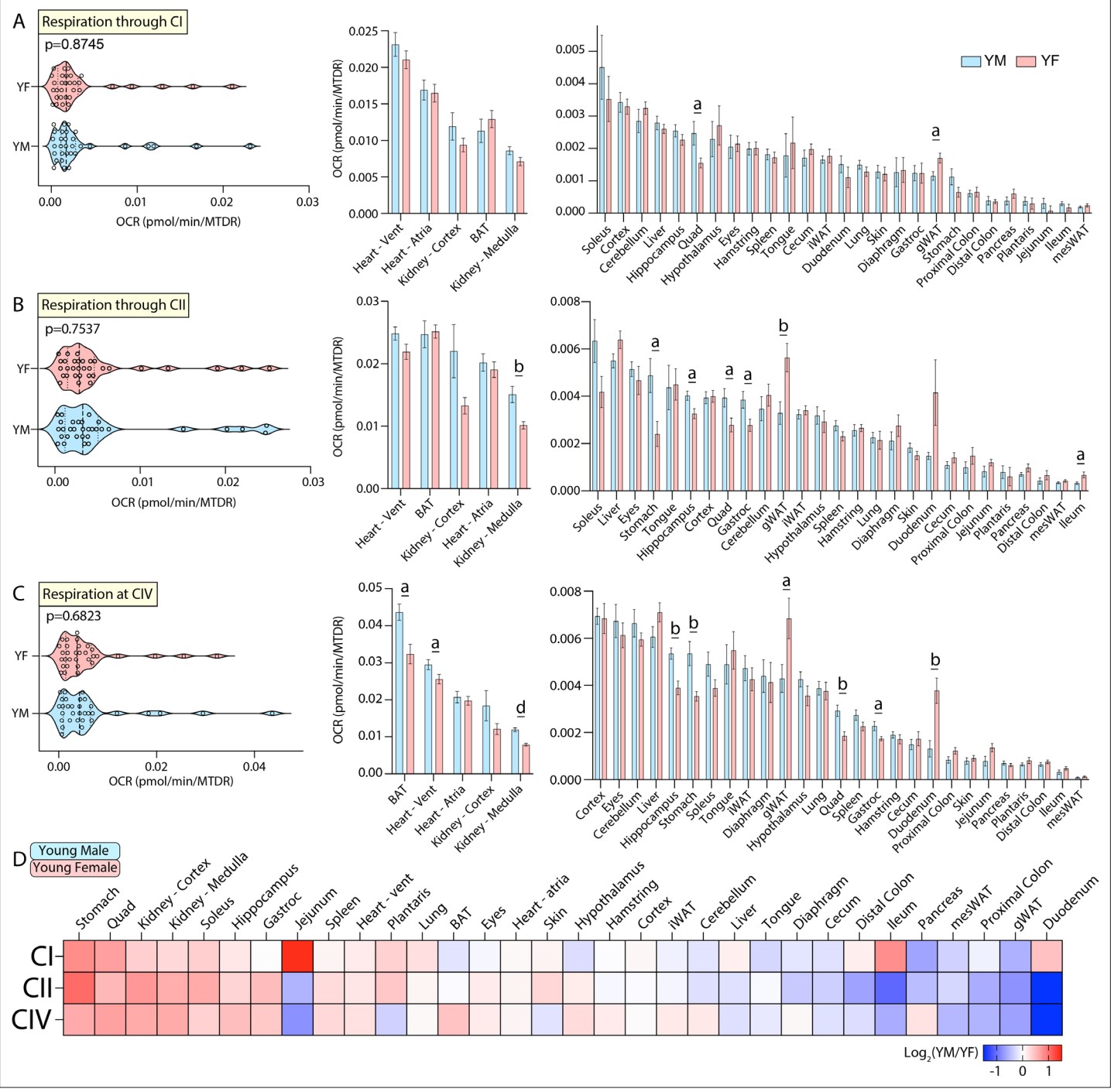

**Figure 3.** Tissue-by-tissue analysis of young male and female mitochondrial function. (**A**) (Left) Systems-level view of mitochondrial respiration (NADH-stimulated) through complex I (CI). (Right) Mitochondrial respiration through CI across all young tissues. (**B**) (Left) Systems-level view of mitochondrial respiration through complex II (CII, succinate-stimulated in the presence of rotenone to inhibit CI). (Right) Mitochondrial respiration through CII across all tissues. (**C**) (Left) Systems-level view of mitochondrial respiration at complex IV (CIV) in the presence of rotenone and antimycin A to inhibit CI and CIII, respectively. (Right) Mitochondrial respiration at CIV across all tissues. All data is presented as the mean with standard error, and organized highest to lowest for young male values. (**D**) Heat map view of mitochondrial function across all tissues assayed (omitting reproductive organs). Data is represented as young male/female. Tissues with elevated respiration in males appear red while the same for females appear blue. Data is organized highest to lowest by summation of young male CI, CII, and CIV respiration values. n=10 young male (YM, 10 weeks) and 10 young female (YF, 10 weeks) per tissue assayed. All oxygen consumption rates (OCR) are normalized to mitochondrial content (based on MTDR). Statistical significance is represented as: a=p<0.05, b=p<0.01, c=p<0.001, and d=p<0.0001. BAT, brown adipose tissue; gWAT, gonadal white adipose tissue; iWAT, inguinal white adipose tissue; Quad, quadriceps muscles; Vent, ventricles.

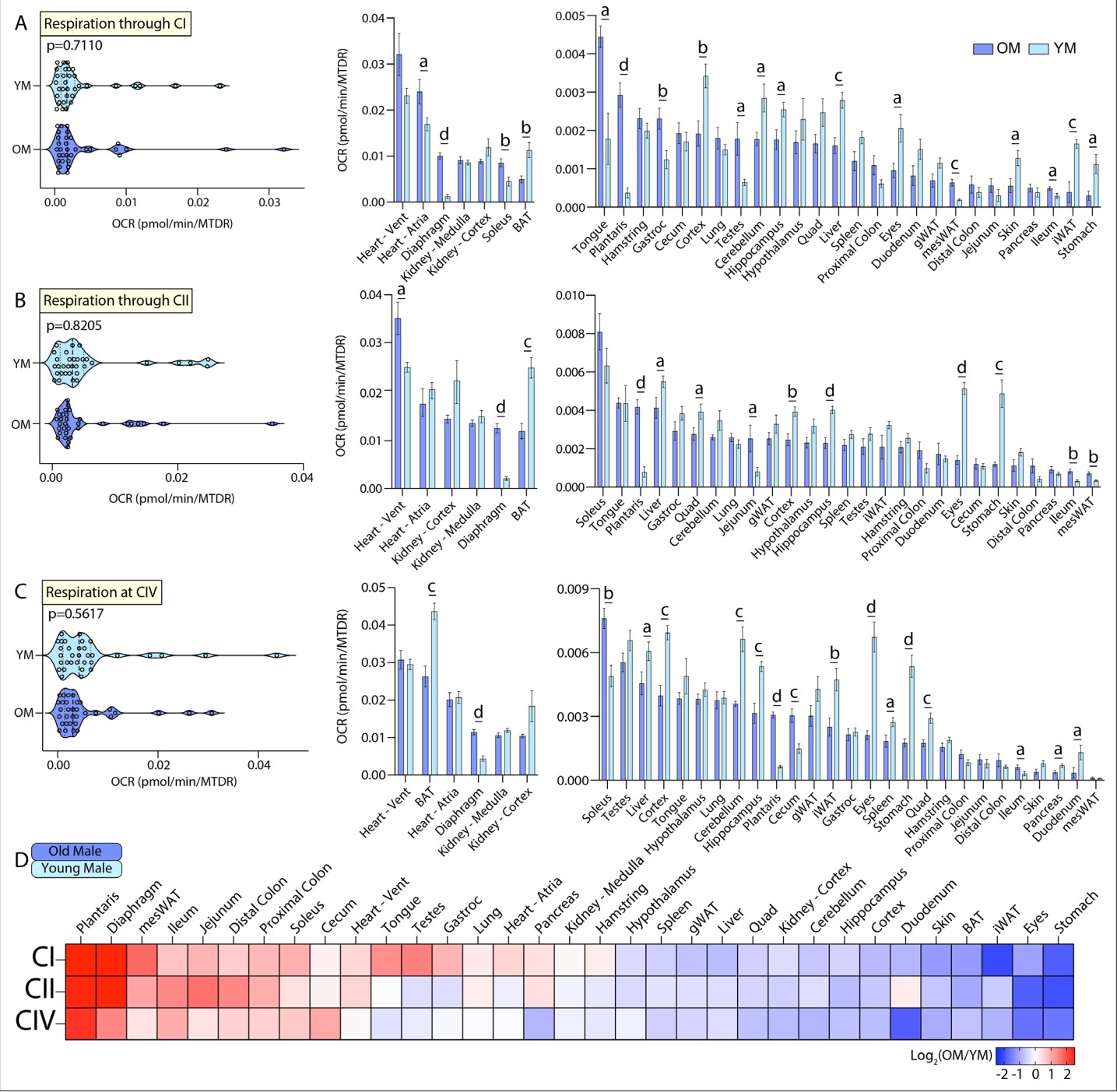

**Figure 4.** Tissue-by-tissue analysis of young and old male mitochondrial function. (**A**) (Left) Systems-level view of mitochondrial respiration (NADH-stimulated) through complex I (CI). (Right) Mitochondrial respiration through CI across all male tissues. (**B**) (Left) Systems-level view of mitochondrial respiration (succinate-stimulated in the presence of rotenone to inhibit CI) through complex II (CII). (Right) Mitochondrial respiration through CII across all male tissues. (**C**) (Left) Systems-level view of mitochondrial respiration at complex IV (CIV) in the presence of rotenone and antimycin A to inhibit CI and CIII, respectively. (Right) Mitochondrial respiration at CIV across all male tissues. All data is presented as the mean with standard error and organized highest to lowest for old male values. (**D**) Heat map view of mitochondrial function across all male tissues assayed. Data is presented as old/young male. Tissues with elevated respiration in old males appear red while the same for young males appear blue. Data is organized highest to lowest by summation of old male CI, CII, and CIV respiration values. n=10 old male (OM, 80 weeks) and 10 young male (YM, 10 weeks) per tissue assayed. All oxygen consumption rates (OCR) are normalized to mitochondrial content (based on MTDR). Statistical significance is represented as: a=p<0.05, b=p<0.01, c=p<0.001, and d=p<0.0001. BAT, brown adipose tissue; gWAT, gonadal white adipose tissue; iWAT, inguinal white adipose tissue; Quad, quadriceps muscles; Vent, ventricles.

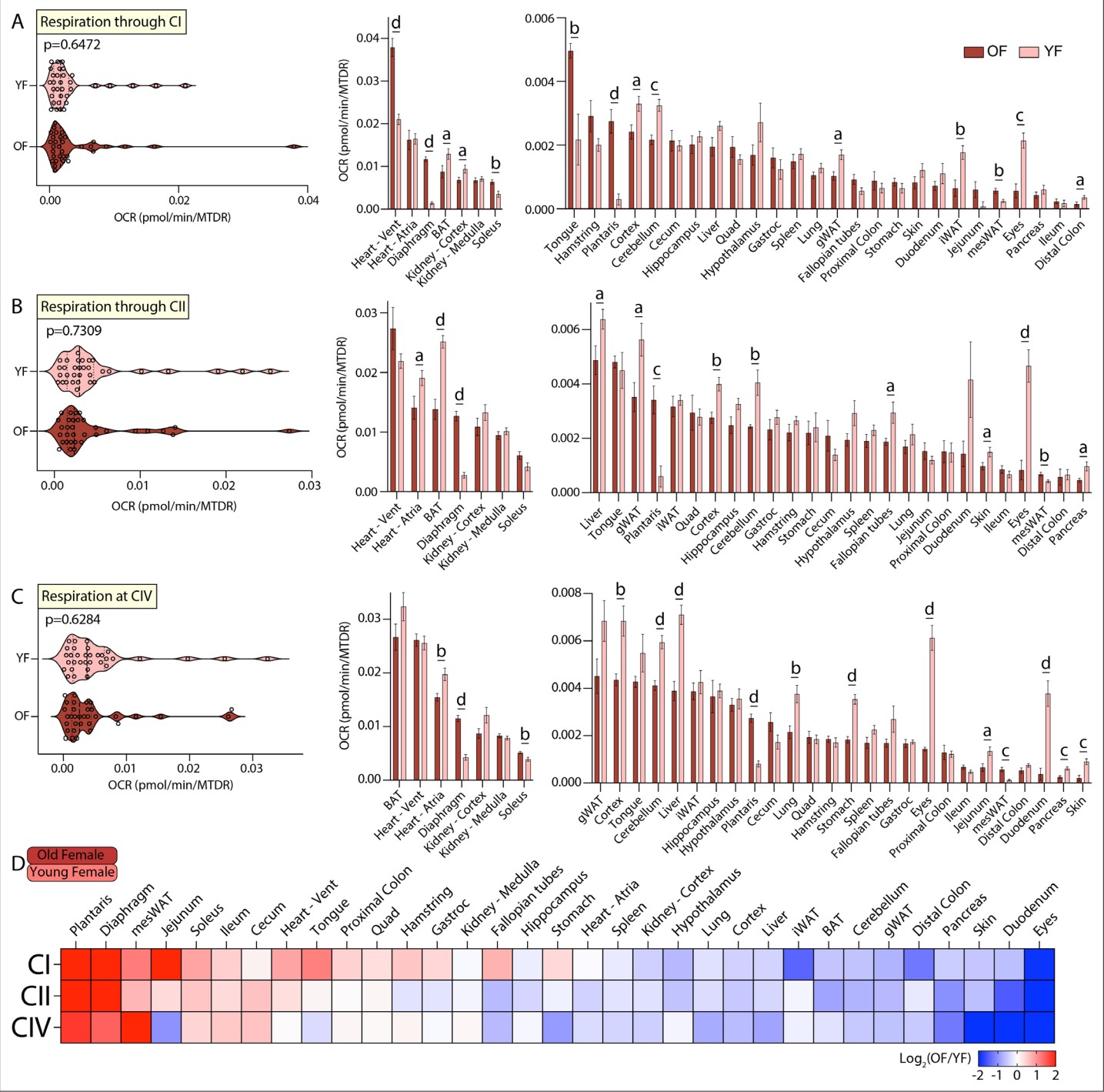

**Figure 5.** Tissue-by-tissue analysis of young and old female mitochondrial function. (**A**) (Left) Systems-level view of mitochondrial respiration (NADH-stimulated) through complex I (CI). (Right) Mitochondrial respiration through CI across all female tissues. (**B**) (Left) Systems-level view of mitochondrial respiration (succinate-stimulated in the presence of rotenone to inhibit CI) through complex II (CII). (Right) Mitochondrial respiration through CII across all female tissues. (**C**) (Left) Systems-level view of mitochondrial respiration at complex IV (CIV) in the presence of rotenone and antimycin A to inhibit CI and CIII, respectively. (Right) Mitochondrial respiration at CIV across all female tissues. All data is organized highest to lowest for old female values. (**D**) Heat map view of mitochondrial function across all female tissues assayed. Data is presented as old/young female, so tissues with elevated mitochondrial respiration in old females appear red while the same for young females appear blue. Data is organized highest to lowest by summation of old female CI, CII, and CIV respiration values. n=10 old female (OF, 80 weeks) and 10 young female (YF, 10 weeks) per tissue assayed. All oxygen consumption rates (OCR) are normalized to mitochondrial content (based on MTDR). Statistical significance is represented as: a=p<0.05, b=p<0.01, c=p<0.001, and d=p<0.0001. BAT, brown adipose tissue; gWAT, gonadal white adipose tissue; iWAT, inguinal white adipose tissue; Quad, quadriceps muscles; Vent, ventricles.

## Age differentially affects mitochondrial function in male and female mice

The fourth tissue-by-tissue comparison of mitochondrial function was made between old male and female mice, where we focused on the effects of sex on mitochondrial respiration in old mice. A systems-level view of respiration through CI, CII, or CIV across all 33 tissues showed no differences in OCR distribution across sex in old mice (*Figure 6A–C*, left panel). At the individual tissue level, however, old male mice had greater mitochondrial activity via CI in the heart atria, kidney (medulla and cortex), lung, and ileum, and lower respiration in the BAT and stomach as compared to old females. Respiration through CII showed that old males have higher mitochondrial activity in the kidney (cortex and medulla), lung, and pancreas, and reduced mitochondrial activity in the stomach, relative to old females (*Figure 6B*, right panel). Respiration at CIV showed that old males have higher mitochondrial respiration in the heart atria, kidney medulla, soleus muscle, lungs, and eyes, and reduced mitochondrial activity in the cerebellum, iWAT, and mesWAT, when compared to old females (*Figure 6C*, right panel). A global and relative view of the sex-specific differences across age showed that old males have elevated mitochondrial activity in the distal colon, lung, eyes, pancreas, soleus, kidney, and heart when compared to old females, whereas old females have higher mitochondrial activity in all adipose tissue depots, stomach, and nearly all brain regions when compared to old males (*Figure 6D*).

## Age has a much larger effect than sex on mitochondrial function

To assess the relative contributions of sex and age to mitochondrial respiration, we first counted the total number of significant differences from all across-group comparisons made in *Figures 3–6*, which was 128 in total. Clustering significant differences by the type of respiration affected showed CIV to have the most with 49 differences, followed by CI with 41 differences, and CII with 38 differences (*Figure 7A*). Grouping significant differences by the across-group comparison in which they occurred showed the young male (YM)-by-old male (OM) comparison to have the most significant differences with 48 in total, followed by young female (YF)-by-old female (OF) with 42, OM-by-OF with 20, and YM-by-YF with 18 significant differences (*Figure 7B*). Clustering the significant difference counts by type, sex (originating from YM-by-YF or OM-by-OF), or age (originating from YM-by-OM or YF-by-OF), we saw that although both sex and age exerted an effect on mitochondrial respiration, it was age that resulted in the highest number of significant changes (*Figure 7B*). We next sought to quantify the number of tissues significantly affected in at least one mitochondrial parameter per across-group comparison. There were 10 tissues with a significant difference when comparing young male and female, 13 when comparing old male and female, 26 tissues when comparing young and old male, and 21 for young and old female (*Figure 7C*). Together, these data indicate that the majority of significant differences observed are the result of age.

Next, we calculated the absolute difference of means and summed the resultant values based on their effect-type, sex (originating from YM-by-YF or OM-by-OF) or age (originating from YM-by-YF or OM-by-OF) (*Figure 7D*). This analysis offered an added magnitude perspective to our view of changes in mitochondrial respiration across sex and age. It allowed us to see that respiration via CI, CII, and CIV all have a greater magnitude of difference as a result of age as compared to sex. Each graph in *Figure 7D* shows ranked values (lowest/left to highest/right) within the age and sex groupings with the top tissues colored and labeled with their percent contribution to the total. The highest contributors to the differences in respiration via CI as a result of age were heart (ventricles and atria), BAT, and diaphragm muscle, while those for sex were heart (ventricles and atria) and BAT. The highest contributors to the differences in respiration through CII as a result of age were BAT, diaphragm muscle, heart ventricles, and kidney cortex, while those for sex were kidney (medulla and cortex) and heart (ventricles and atria). The highest contributors to the differences in respiration at CIV as a result of age were BAT, diaphragm, kidney cortex, and eyes, while those for sex were BAT and heart ventricles, and kidney (medulla and cortex). It is important to note that these rankings do not indicate directionality of a difference, only total magnitude. Additionally, these changes are represented as the absolute magnitude of difference. Therefore, tissues with lower absolute oxygen consumption values would skew toward lower ranking, even though the relative change occurring within that tissue may be large.

Principal component (PC) analysis of each mitochondrial respiration dataset showed a primary and consistent separation across PC1 as a result of age (*Figure 7E*). A small percentage of the variance can be explained through PC2 as a result of sex, although this effect was not as consistent as that seen

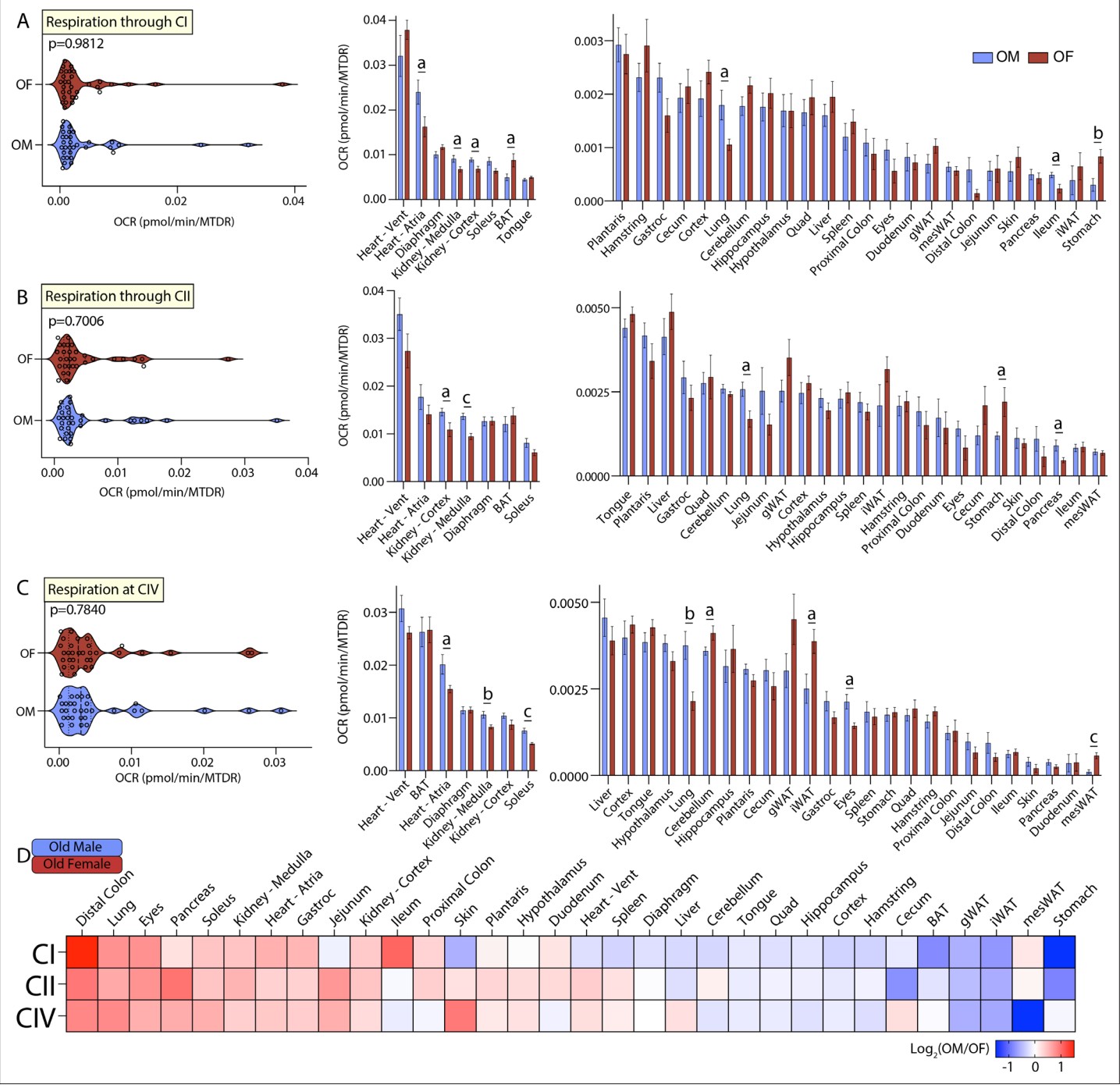

**Figure 6.** Tissue-by-tissue analysis of old male and female mitochondrial function. (**A**) (Left) Systems-level view of mitochondrial respiration (NADH-stimulated) through complex I (CI). (Right) Mitochondrial respiration through CI across all old tissues. (**B**) (Left) Systems-level view of mitochondrial respiration (succinate-stimulated in the presence of rotenone to inhibit CI) through complex II (CII). (Right) Mitochondrial respiration through CII across all tissues. (**C**) (Left) Systems-level view of mitochondrial respiration at complex IV (CIV) in the presence of rotenone and antimycin A to inhibit CI and CIII, respectively. (Right) Mitochondrial respiration at CIV across all tissues. All data is presented as the mean with standard error and organized highest to lowest for male values. (**D**) Heat map view of mitochondrial function across all tissues assayed. Data is represented as old male/female, so tissues with elevated mitochondrial respiration in males appear red while the same for females appear blue. Data is organized highest to lowest by summation of male CI, CII, and CIV respiration values. n=10 old male (OM, 80 weeks) and 10 old female (OF, 80 weeks) per tissue assayed. All oxygen consumption rates (OCR) are normalized to mitochondrial content (based on MitoTracker Deep Red [MTDR]). Statistical significance is represented as: a=p<0.05, b=p<0.01, c=p<0.001, and d=p<0.0001. BAT, brown adipose tissue; gWAT, gonadal white adipose tissue; iWAT, inguinal white adipose tissue; Quad, quadriceps muscles; Vent, ventricles.

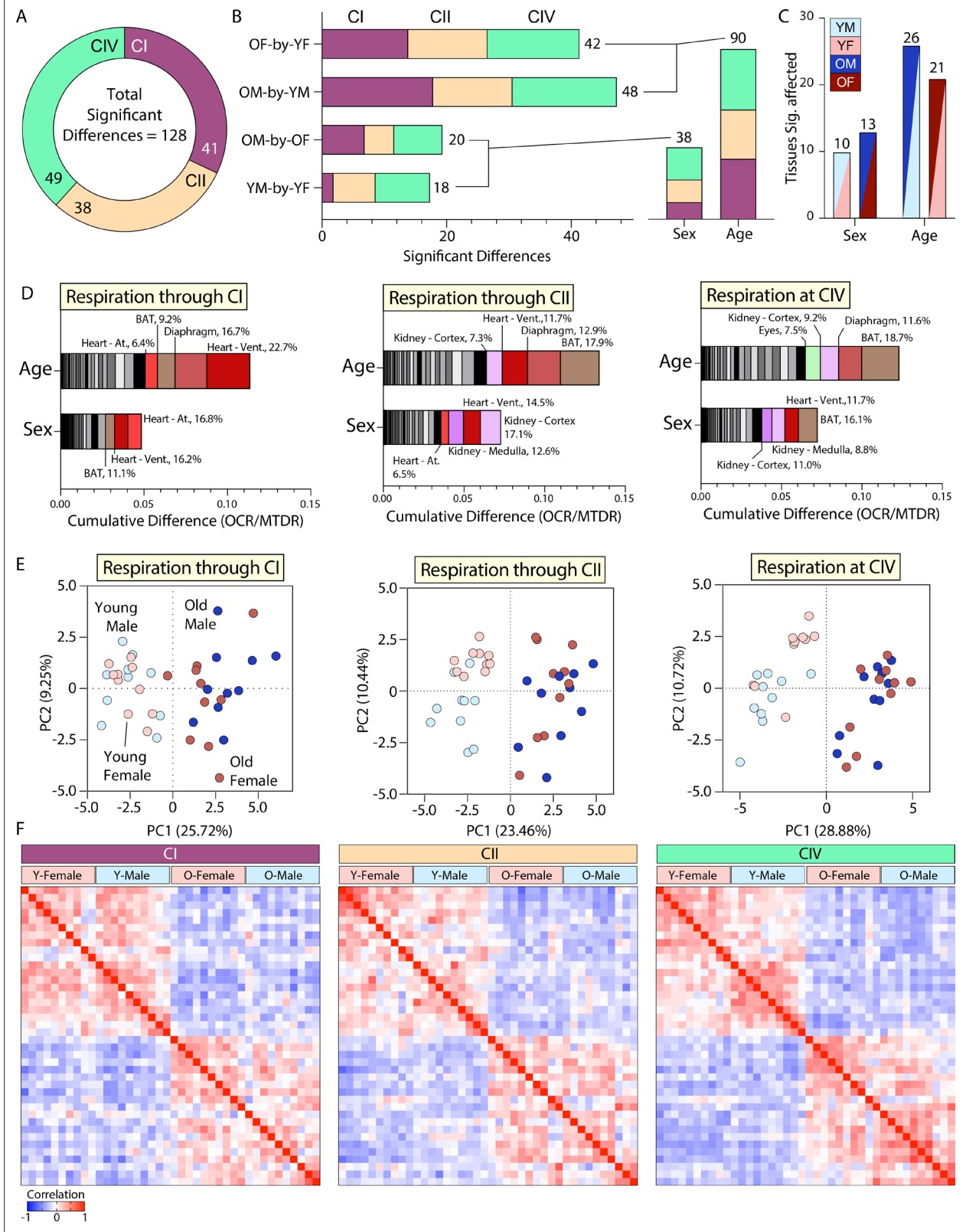

**Figure 7.** Mitochondrial respiration is affected by sex but dominated by age. (**A**) Total number of statistically significant differences across all tissue-by-tissue comparisons, grouped by the component measured (CI = respiration through complex I, CII = respiration through complex II, or CIV = respiration at complex IV). (**B**) Total number of significant differences across tissue-by-tissue comparisons grouped by the specific comparison, colored to represent the mitochondrial component in which respiration began (CI, CII, or CIV), and summed to the right of each histogram to highlight the

*Figure 7 continued on next page*

*Figure 7 continued*

number of significant findings per comparison. To underscore the number of statistically significant sex- or age-associated differences, the total number of significant findings from YM-by-YF and OM-by-OF (sex effect), and OM-by-YM and OF-by-YF (age effect) were summed. (**C**) Quantification of the total number of tissues affected in each tissue-by-tissue comparison. Data are colored based on the specific comparison and grouped as sex- or age-associated to illustrate the effect-type. (**D**) Cumulative absolute difference of means for CI, CII, and CIV (from left to right) grouped by effect-type, sex (originating from YM-by-YF or OM-by-OF) or age (originating from OM-by-YM or OF-by-YF). These graphs do not indicate directionality of the change, only the absolute cumulative magnitude. Each box within the histogram represents a unique tissue. All histograms are organized lowest (left) to highest (right) in degree of tissue contribution to total change given the comparison type. The top contributors to sex- or age-associated changes are highlighted with non-grayscale colors and their relative percentage of contribution to the total cumulative difference is provided. (**E**) Principal component analysis (PCA) of all tissues and groups for CI, CII, and CIV (from left to right). (**F**) Pearson correlation heat maps of all tissues combined from young and old, male and female mice. From top to bottom and left to right the samples are organized by group as follows: YF, YM, OF, and OM (n=10 mice per group). YM = young male, YF = young female, OM = old male, OF = old female. Young = 10 weeks, old = 80 weeks. BAT, brown adipose tissue; gWAT, gonadal white adipose tissue; iWAT, inguinal white adipose tissue; Quad, quadriceps muscles; At, atria; Vent, ventricles.

with age. Pearson correlation heat maps further highlighted age as a key factor relating samples across groups (*Figure 7F*). Male and female samples of the same age positively correlated with one another, while young and old samples regardless of sex showed a negative correlation. These combined data indicate that while sex does affect mitochondrial respiration, age is the dominant factor.

## The sex-specific impact of age on mitochondrial function

To investigate the sex-dependent effects on mitochondrial function across age, we first calculated all of the relative change values for males and females from young to old. Summation of relative change values of all tissues within a specific mitochondrial parameter - respiration through CI, CII, or CIV - allowed us to view sex-specific systems-level effects across age (*Figure 8A and B*). Both the male and female systems showed a relative net increase in CI and decrease in CIV activity with age. Interestingly, with respect to CII activity, the male and female systems had divergent responses. The net relative CII activity increased in males and decreased in females with age. These data suggest that: (1) independent of sex, CI and CIV are uniquely regulated, showing net opposite responses with age, and (2) CII activity is the most sex-affected respiratory component across age and tissues.

Heat maps of the individual tissue-level data presented in *Figure 8A and B* further highlight the tissue-level similarities and differences across age, sex, and mitochondrial respiration type (i.e. through CI, CII, or CIV). For example, respiration through CI, CII, or CIV showed that the plantaris muscles (PL), diaphragm muscle (DI), jejunum (JE), mesenteric white adipose tissue (MW), and ileum (IL) have the greatest positive relative change in response to age, while the eye (EY), skin (SK), stomach (ST), and BAT (BT) were among the tissues that showed the greatest negative relative change with age (*Figure 8C*).

To determine the tissues with the strongest sex-specific age effects, we organized male and female data by the largest (left) to smallest (right) difference of relative change. This multi-dimensional view highlights differences within a tissue across sex in response to age (*Figure 8D*). The largest sex-specific effects of age on respiration via CI were found in the stomach (ST), jejunum (JE), distal colon (DC), quadriceps muscles (QD), and pancreas (PN); in contrast, the gWAT (GW), kidney cortex (KC), diaphragm (DI), cecum (CC), soleus muscle (SL), and cerebellum (CE) displayed the highest similarity of change (*Figure 8D*, top panel). The largest sex-specific effects of age on respiration through CII were found in the stomach (ST), duodenum (DU), distal colon (DC), pancreas (PN), and jejunum (JE); in contrast, the liver (LV), hamstring muscle (HS), kidney medulla (KM), spleen (SP), and skin (SK) displayed the smallest sex-differences (*Figure 8D*, middle panel). Finally, the most sex-specific effects of age on respiration at CIV were found in mesenteric white adipose tissue (MW), duodenum (DU), jejunum (JE), skin (SK), and distal colon (DC); in contrast, the tongue (TN), gastrocnemius muscle (GS), heart ventricles (HV), hypothalamus (HY), and diaphragm (DI) displayed the greatest similarity of change with age (*Figure 8D*, bottom panel).

All sexually divergent tissues are marked specifically within each graph per mitochondrial respiration-type. Respiration via CI showed seven tissues with sexual divergence in response to aging, and these were stomach (ST), distal colon (DC), pancreas (PN), lung (LN), heart atria (HA), quadriceps muscle complex (QD), and kidney medulla (KM) (*Figure 8D*, top panel). Respiration through CII showed five tissues with sexual divergence in response to aging, and these were duodenum (DU), pancreas (PN), lung (LN), quadriceps muscle complex (QD), and distal colon (DC) (*Figure 8D*, second panel).

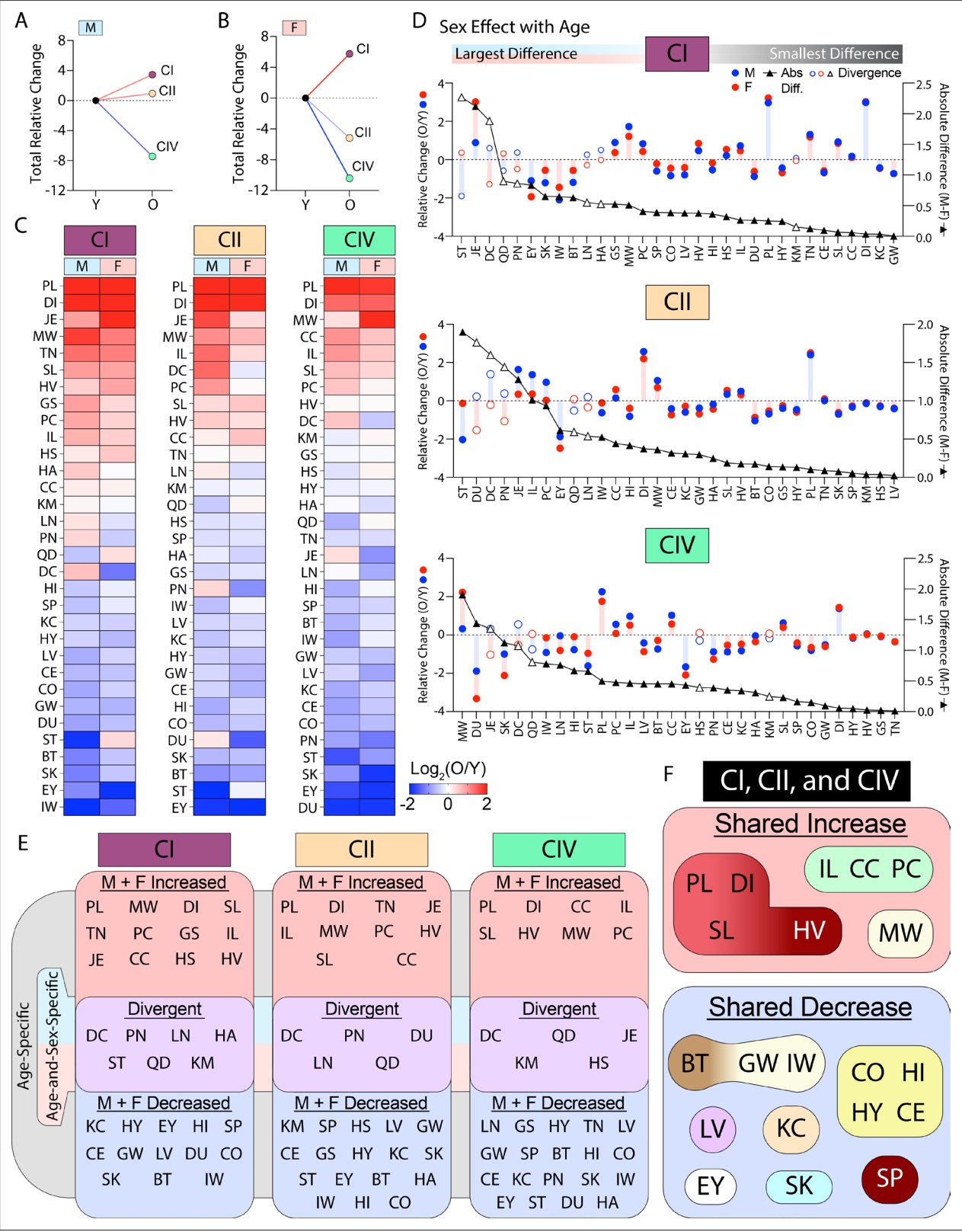

**Figure 8.** The effects of age on mitochondrial respiration occur in sex-specific ways. (**A, B**) Systems-level view of the total relative change (sum of all relative changes) present at each mitochondrial parameter (respiration via CI, CII, or CIV) for males (M) and females (F), respectively. All change is old (O) compared to young (Y). (**C**) Heat maps of O/Y data for each mitochondrial parameter per tissue across the lifespan. Male and female values are grouped and organized by greatest (top) to least (bottom) sum of relative change per tissue. Tissues with the largest positive relative change as a result

*Figure 8 continued on next page*

*Figure 8 continued*

of aging are located on the top-most region of the heat maps, while those with the largest negative relative change are at the bottom. The middle of each heat map represents tissues with little relative response to aging or with opposite effect directions across sex. (**D**) Graphs showing magnitude of sex-effect with age across all tissues and mitochondrial parameters (CI, CII, or CIV). The blue line and circles represent $\log_2$(old male/young male) data and the red line and boxes represent $\log_2$(old female/young female) data; both linked to the left y-axis. The black line and triangles represent the absolute difference of male and female relative change data (linked to the right y-axis). All data is organized from left to right by highest to lowest absolute difference of relative change across sex. Tissues with the greatest relative difference across sex are located to the left-most region of the graph, while those with the most similar response to aging are located to the right. A hollow blue circle, red circle, and black triangle represent a tissue with a divergent sex-specific age response. (**E**) Summary diagrams classifying the relative trends of each mitochondrial parameter assayed per tissue as age- or age-and-sex-specific. Age-specific effects have a shared relative change direction across sex. Tissues in the red 'M+F Increased' box have a positive relative change for both males and females. Tissues in the blue 'M+F Decreased' box have a negative relative change for both males and females. Tissues in the purple box labeled 'Divergent' have opposing $\log_2$(O/Y) directions (signs, + or -) for male and female values; these tissues display relative mitochondrial changes that are age-and-sex-specific. (**F**) Summary diagram showing tissues with a shared increase (red box) or decrease (blue box) consistent across all complexes (CI, CII, and CIV). Tissues within the shared increase or decrease classifications are grouped by their tissue-type, for example. PL, DI, SL, and HV are all muscle, CO, HI, HY, and CE are all brain and so on. CI = respiration measured through complex I stimulated by NADH; CII = respiration through complex II stimulated by succinate in the presence of rotenone; CIV = respiration at complex IV stimulated by TMPD (*N,N,N',N'*-tetramethyl-*p*-phenylenediamine) and ascorbate in the presence of rotenone and antimycin A. Tissues: BT = brown adipose tissue, CC = cecum, CE = cerebellum, CO = brain cortex, DI = diaphragm, DC = distal colon, DU = duodenum, EY = eyes, GS = gastrocnemius, GW = gonadal white adipose tissue, HS = hamstring, HA = heart atria, HV = heart ventricles, HI = hippocampus, HY = hypothalamus, IL = ileum, IW = inguinal white adipose tissue, JE = jejunum, KC = kidney cortex, KM = kidney medulla, LV = liver, LN = lung, MW = mesenteric white adipose tissue, PN = pancreas, PL = plantaris, PC = proximal colon, QD = quadriceps, SK = skin, SL = soleus, SP = spleen, ST = stomach, TN = tongue. Reproductive organs are omitted from cross-sex analysis.

Respiration at CIV also showed five tissues with sexual divergence in response to aging, and these were jejunum (JE), distal colon (DC), hamstring (HS), quadriceps muscle complex (QD), and kidney medulla (KM) (*Figure 8D*, bottom). Together, these data show the relative magnitude of change across sex and age, and highlight the tissue- and sex-specific regulation of mitochondrial respiration with aging.

We further grouped the above data by directionality to visualize male and female tissue responses to aging (*Figure 8E*). From this, we can quickly visualize tissues displaying a shared direction of change in respiration via CI, CII, or CIV across sex and age. Additionally, some tissues responded in an age-and-sex-specific manner (i.e. display divergence in *Figure 8D*). This data clustering summarizes the shared (age-specific) or opposite (sex-specific) directionality of mitochondrial function across tissues in response to age.

It should be noted, however, that the mitochondrial activity of a particular tissue could increase or decrease in a sex-specific way even though the directionality is the same. For example, respiration through CII for the stomach (ST) showed male and female values decreasing with age, yet the male had a larger relative decrease when compared to female (*Figure 8D*, middle panel). Another consideration should be given to the divergent group, which displayed varying degrees of divergence. For example, CI-associated respiration of the stomach (*Figure 8D*, top panel) showed a strong divergence given that male had a strong decline in function while the female increased over time. On the other hand, the CI-associated respiration of the kidney medulla showed a weak divergence; although the directionality of change was opposite, the magnitude was small.

Lastly, we grouped all the tissues with a consistent shared (across sex) relative increase or decrease in CI-, CII-, and CIV-associated respiration in response to age (*Figure 8F*). The tissues with a shared (across sex) increased response to aging were plantaris (PL), diaphragm (DI), soleus (SL), heart ventricle (HV), ileum (IL), cecum (CC), proximal colon (PC), and mesenteric fat (MW). Interestingly, PL, DI, and SL are all skeletal muscles and IL, CC, and PC are all part of the digestive tract. Also of note, mitochondrial respiration in the gastrocnemius (GT), hamstring (HS), and tongue (TN) muscles were not uniformly increasing with age. In fact, these muscles showed reduced mitochondrial respiration with age, in striking contrast to other skeletal muscle types assayed. The tissues with a shared (across sex) decrease in response to aging were brown fat (BT), gonadal fat (GW), inguinal fat (IW), brain cortex (CO), hippocampus (HI), hypothalamus (HY), cerebellum (CE), liver (LV), kidney cortex (KC), eye (EY), skin (SK), and spleen (SP). Different fat depots and most of the brain regions showed a consistent decline in respiratory function with age, regardless of sex. The kidney cortex (KC) oddly showed a respiratory effect distinct from the medulla, which was almost completely resistant to age-related respiratory changes. The eyes (EY) not only showed a consistent decline in respiration with age, but

also had one of the largest combined relative effects. Together, the shared and divergent analyses presented (*Figure 8E and F*) illuminate the modulation of mitochondrial function across age.

## Discussion

Here, we provide the most comprehensive catalog of sex- and age-associated mitochondrial respiration signatures across diverse tissues and organs system. One important aspect of our study to emphasize is the experimental uniformity. All the tissue samples were collected from the same set of mice, and processed and analyzed in a uniform and standardized manner on the same respirometry platform in a single laboratory. Combined with the 10 biological replicates, our experimental setup and workflow markedly enhanced the rigor and reduced the variations in data quality. From the mitochondrial respiration compendium, we can clearly see that mitochondrial function varies widely across tissues. This is perhaps not surprising given the highly variable energetic demands of different tissues (*Rolfe and Brown, 1997*) and the known differences in the composition of mitochondrial proteomes across tissues (*Mootha et al., 2003*; *Pagliarini et al., 2008*). Importantly, our data provide evidence that aging has a disproportionately larger effect than sex on mitochondrial activity across tissues.

The transcriptomic dataset generated by the Tabula Muris Consortium clearly indicates organ-specific temporal signatures across the lifespan (*Schaum et al., 2020*; *Zhang et al., 2021*; *Almanzar et al., 2020*), and this is assumed to track closely with tissue function even though no equivalent systematic interrogation of tissue function across the lifespan has been performed. In this regard, our mitochondrial activity atlas aligned and corroborated with the conclusion from large-scale transcriptomic analyses that the aging process and trajectory can vary substantially across tissues (*Schaum et al., 2020*; *Zhang et al., 2021*). This reinforces the notion that while an organism ages, the aging-associated decline in organ function is not uniform across tissues and organ systems. Given that genetics and environmental factors all contribute to tissue health and their functional breakdown with age, this further exemplifies the heterogenous process of aging across a population.

Although sex is a significant biological variable influencing many aspects of physiology and pathophysiology (*Mauvais-Jarvis et al., 2020*; *Austad and Fischer, 2016*), and sex hormones are known to affect mitochondrial function (*Klinge, 2020*; *Gaignard et al., 2017*; *Sultanova et al., 2020*), we were surprised that sex has a relatively modest impact on respiratory capacity in mitochondria across tissues and age. The modest impact of sex on mitochondrial respiration across tissues is also concordant with the relatively minor impact of sex on tissue transcriptome across the lifespan (*Schaum et al., 2020*; *Ma et al., 2020*). The caveat, however, is that our study is based on a single strain of mice with a uniform genetic background. The contribution of biological sex may be more pronounced in an outbred population where sex hormones interact with other genetic and environmental determinants to influence mitochondrial activity across tissues and age.

Aging is a dynamic process that unfolds across the lifespan, which involves changes at the molecular, cellular, biochemical, and tissue level. Although the magnitude of change may vary between individuals, with increasing age, humans generally experience a gradual and progressive decline in cognitive function, visual acuity, skin elasticity, motor control, digestive function, fertility, and metabolism (e.g. increased adiposity) (*López-Otín et al., 2013*; *Tian et al., 2023*). Interestingly, in our dataset, we also observed an aging-associated decline in mitochondrial respiration in most brain regions, eyes, skin, fallopian tubes, different fat depots and BAT, pancreas, and the digestive tracts in male and/or female mice. This age-dependent reduction in mitochondrial function likely contributes, at least in part, to the functional decline across different organ systems. Depending on the tissue, the magnitude of change in mitochondrial respiration in old mice, when compared to young mice, can be substantial or relatively modest. This variability is consistent with the heterogenous rate of aging across tissues (*Tian et al., 2023*).

The underlying cause of reduced mitochondrial respiration in aging is not well understood and it may involve multiple mechanisms. Transcriptomic analyses have consistently shown that aging induces a reduction in mitochondrial OXPHOS gene expression (*Schaum et al., 2020*; *Zhang et al., 2021*), and this likely is one of the mechanisms that contributes to the reduced mitochondrial activity we observed across many tissues in old mice. The contributing factors which drive an age-dependent decrease in mitochondrial OXPHOS gene expression across tissues, however, is largely unknown. Other potential mechanisms may involve age-dependent accumulation of oxidative damage in mitochondrial DNA, proteins, and lipids that adversely affect mitochondrial function (*Sun et al., 2016*;

*Sastre et al., 2000*). The damage to mitochondrial integrity in aging is frequently attributed to reactive oxygen species (ROS) that are naturally generated as byproducts of electron transfer through the mitochondrial OXPHOS complexes (*Orrenius et al., 2007*), or to inadequate stress response pathways (e.g. proteostasis and mitophagy) (*Lima et al., 2022*). While cells have mechanisms to guard against oxidative stress due to mitochondrial activity, these antioxidant defense mechanisms are known to decline with age despite an age-dependent increase in ROS production (*Jones et al., 2002*; *Kozakiewicz et al., 2019*). Similarly, the integrated stress response pathways also tend to diminished with age (*Lima et al., 2022*). Thus, it is often thought the gradual accumulation of ROS with age, coupled with a decline in mitochondria's ability to handle proteotoxic stress, progressively compromises mitochondrial function.

It should be noted that changes in cell, tissue, and organ function are often counteracted by homeostatic and compensatory responses. While many tissues showed a significant reduction in mitochondrial function, some tissues also showed a concomitant increase in mitochondrial activity with age. For example, we observed an increase in mitochondrial respiration in the heart, different skeletal muscle groups, and mesenteric fat in old male and/or female mice. Increased mitochondrial respiration is seen in both the oxidative (diaphragm, soleus, tongue) and glycolytic (plantaris) muscle fibers. This increase in mitochondrial activity is not attributed to differences in mitochondrial content since the OCR is normalized to mitochondrial content.

Skeletal muscle mass is known to decrease with age (*Wilkinson et al., 2018*), and it is frequently but not always accompanied by a reduction in skeletal muscle mitochondrial function in vivo (*Gouspillou et al., 2014*; *Short et al., 2005*; *Conley et al., 2000*; *Kent-Braun and Ng, 2000*; *Chilibeck et al., 1998*) and in vitro (*Short et al., 2005*; *Mansouri et al., 2006*; *Kumaran et al., 2005*; *Tonkonogi et al., 2003*; *Drew et al., 2003*; *Picard et al., 2010*). Some studies, however, have also suggested that the energetic efficiency of mitochondria in skeletal muscle - located beneath the sarcolemmal membrane or between the myofibrils - appears to increase with age (*Crescenzo et al., 2015*; *Crescenzo et al., 2014*). While we observed an increase in the maximal uncoupled respiration in skeletal muscle lysates of aged mice, our results cannot be directly compared to prior studies that examine coupled respiration in intact tissue or isolated mitochondria. It has been shown that skeletal muscle mitochondrial number increases, whereas mitochondrial size decreases, with age (*Del Campo et al., 2018*). Additionally, the ultrastructure of mitochondria may also change with age (*Brandt et al., 2017*). Whether these parameters correlate with and contribute to age-dependent changes in mitochondrial respiration remains to be established.

Similar to skeletal muscle, the heart also experiences functional decline with age, and this is thought to be attributed to multiple mechanisms that include age-dependent changes in mitochondrial content, morphology, lipid composition (e.g. cardiolipin) and dynamics (e.g. fusion and fission), activity of the electron transport chain, ROS generation, and susceptibility to mitochondrial permeability transition pore opening (*Lesnefsky et al., 2016*; *Boengler et al., 2017*). Additional contributing factors to cardiac functional decline include age-related oxidative stress and chronic low-grade inflammation, which further impact mitochondrial integrity and function (*de Almeida et al., 2020*). With regard to impaired mitochondrial OXPHOS in the aging heart, it appears the subpopulation of mitochondria located between the myofibrils, the interfibrillar mitochondria (IFM), are affected to a much greater extent than the population of mitochondria located in the subsarcolemmal membrane (*Suh et al., 2003*; *Fannin et al., 1999*; *Lesnefsky et al., 2001*). This suggests that IFM, being more closely associated with energy-demanding myofibrils, might be more vulnerable to age-related dysfunction. In contrast to studies using isolated mitochondria from the heart, mitochondrial function was found to be largely preserved in permeabilized intact cardiomyocyte bundles from senescent rats, in which all intact mitochondria are represented (*Picard et al., 2012*).

In our assay, however, maximal mitochondrial respiration in the heart ventricles increases with age. Since the contractile property of the heart declines and the stiffness of the extracellular matrix increases with age due to fibrosis (*Biernacka and Frangogiannis, 2011*; *Travers et al., 2016*), the heart tissue becomes less efficient and requires more energy to carry out its normal pumping function. Thus, we speculate that reduced efficiency of the aging heart may drive a compensatory response to increase mitochondrial capacity. An increase in mitochondrial respiration, however, may come at the cost of generating greater amounts of reactive oxygen species, which are thought to be detrimental to the aging heart. Thus, with the general observation that organ function declines with

age, our data suggests that some tissues such as the heart and skeletal muscle may increase their mitochondrial respiration to meet the energetic demands resulting from reduced mass (e.g. sarcopenia) and/or declining efficiency of organ function. This hypothesis awaits future experimental verification.

It is worth noting that our respirometry analysis assesses NADH-dependent respiration *through* CI, succinate-dependent respiration *through* CII, and TMPD/ascorbate-dependent respiration *at* CIV. Regardless of whether electrons enter the respiratory chain through CI or CII, or at CIV, oxygen (the final electron acceptor) is only consumed at CIV. Since the OCR is normalized to mitochondrial content, the variations we observed in OCR within any given tissue through CI or CII, or at CIV, likely reflect how tightly coupled respiratory complex chains consisting of CI+CIII2+CIV and CII+CIII2+CIV are relative to CIV alone. Although our data suggest that sex and age affect the flow of electrons through respiratory complex CI+CIII2+CIV and CII+CIII2+CIV across tissues, this remains speculative given that our assays were not performed on intact mitochondria.

Every respirometry method has its advantages and limitations (*Salabei et al., 2014*). Unlike the mitochondrial respiration analysis in intact cells or isolated mitochondria, our respirometry analyses were performed in mitochondria-enriched lysates derived from frozen tissues (*Acin-Perez et al., 2020*). In an intact mitochondrion, the rate of respiration is regulated by both substrate supply across the mitochondrial membrane and the rate of NADH and FADH production via the TCA cycle and fatty acid β-oxidation. In the absence of an intact mitochondrial membrane, our assays measure maximal mitochondrial respiration through CI, CII, or CIV. Thus, our mitochondrial activity reflects maximal respiration across tissues. In an in vivo milieu, mitochondrial activity is likely regulated and may not reach its maximal capacity as reflected in our assay. Other relevant and informative parameters concerning mitochondrial functions - e.g., P/O ratios, basal respiration, and ATP-coupled respiration - could not be obtained from our current approach, which relied on non-intact mitochondria. However, despite these limitations, the major advantage of the frozen-tissue method is that it is now feasible to carry out large-scale respirometry analyses across tissues, thus providing a broader understanding of systemic bioenergetic health that outweighs its limitations.

It should be emphasized that our mitochondrial respiration represents the average values for each tissue, even though all tissues consist of multiple distinct cell types that may vary in their cellular composition and mitochondrial activity across sex and age. While individual cell types can be isolated prior to respirometry analysis, this labor-intensive method is not suited for assessing mitochondrial respiration at scale (i.e. across a large number of tissues from the same animal). In the future, this challenge may be overcome with the development of new technologies that allow for high-resolution mitochondrial respirometry at cell-type or single-cell resolution in situ.

In summary, our mitochondria functional signatures demonstrate similar and divergent responses to aging across tissues. Whether changes in tissue mitochondrial respiration reflect the cause, consequence, or both, of aging remains to be determined. We anticipate that the integration of this knowledge and approach with other large-scale omics data will help uncover potential genetic, epigenetic, and biochemical mechanisms linking mitochondrial health and organismal aging.

# Materials and methods
## Mouse model

All wild-type C57BL/6J male and female mice were purchased from the Jackson Laboratory and fed a standard chow (Envigo; 2018SX). In total, the young group was comprised of 10 male and 10 female 10-week-old mice, and the old group was comprised of 10 male and 10 female 80-week-old mice. Mice were housed in polycarbonate cages on a 12 hr:12 hr light-dark photocycle with ad libitum access to water and food. All mice were fasted for 2 hr prior to euthanasia and dissection. Tissues were collected, snap-frozen in liquid nitrogen, and kept at –80°C until analysis. All mouse protocols were approved by the Institutional Animal Care and Use Committee of the Johns Hopkins University School of Medicine (animal protocol # MO22M367). All animal experiments were conducted in accordance with the National Institute of Health guidelines and followed the standards established by the Animal Welfare Acts.

**Table 1.** Assay parameters for Seahorse-based respirometry analysis across all tissues.

Table shows tissues used, approximate size per tissue homogenized for analysis, amount of 1× MAS buffer used for homogenization, and the amount of protein used for respiration analysis across all tissues.

| Tissue | Approximate size | 1× MAS buffer vol (mL) | Protein used (µg/well) |
|---|---|---|---|
| Adipose - BAT | Both sides - entire | 2 | 2 |
| Adipose - gWAT | Both sides - entire | 2 | 15 |
| Adipose - iWAT | Both sides - entire | 2 | 15 |
| Adipose - mesWAT | 100–200 mg | 2 | 15 |
| Brain - Cerebellum | Entire | 2 | 6 |
| Brain - Cortex | Both sides - 25 mg | 2 | 6 |
| Brain - Hippocampus | Both sides - entire | 1 | 6 |
| Brain - Hypothalamus | Entire | 0.75 | 6 |
| Eye | Both sides - entire | 2 | 10 |
| GI - Cecum | Entire | 2 | 10 |
| GI - Large intestine - Distal colon | Entire | 2 | 10 |
| GI - Large intestine - Proximal colon | Entire | 2 | 10 |
| GI - Small intestine - duodenum | Entire | 2 | 10 |
| GI - Small intestine - Ileum | Entire | 2 | 10 |
| GI - Small intestine - Jejunum | Entire | 2 | 10 |
| GI - Stomach | Entire | 2 | 8 |
| Heart - Atria | Both sides - entire | 1.5 | 2 |
| Heart - Ventricle | Both sides - entire | 3 | 2 |
| Kidney - Cortex | 50–100 mg | 3 | 6 |
| Kidney - Medulla | 50–100 mg | 3 | 6 |
| Liver | 50–100 mg | 3 | 8 |
| Lung | Both sides - entire | 3 | 10 |
| Pancreas | Entire | 1 | 8 |
| Sex - Fallopian tubes | Entire | 2 | 10 |
| Sex - Testes | Single testes - entire | 2 | 8 |
| Skeletal muscle - Diaphragm | Entire | 1 | 10 |
| Skeletal muscle - Gastrocnemius | Both sides - entire | 3 | 10 |
| Skeletal muscle - Hamstring | Both sides - entire | 3 | 8 |
| Skeletal muscle - Plantaris | Both sides - entire | 1 | 10 |
| Skeletal muscle - Quadriceps | Both sides - entire | 3 | 10 |
| Skeletal muscle - Soleus | Both sides - entire | 1 | 8 |
| Skeletal muscle - Tongue | Entire | 1.5 | 8 |
| Skin | 100–200 mg | 1 | 10 |
| Spleen | Entire | 2 | 8 |

## Comprehensive multi-organ dissection

Each mouse was dissected cleanly, swiftly, and in a concerted manner by three people. Each dissection took approximately 8–10 min, with 33 tissues collected per dissection. After euthanasia, blood was collected via decapitation. The head was then immediately given to dissector one to collect brain regions (hypothalamus, cerebellum, hippocampus, and cortex), eyes, and tongue. Simultaneously, the visceral cavity was opened by dissector two. iWAT was collected immediately after opening the abdominal skin. After that the abdominal muscle was cut and gWAT and testes or fallopian tubes were collected. Following this, the visceral organs were partitioned in two groups. Dissector two promptly dissected liver, stomach, kidneys (further separated the cortex and medulla), spleen, diaphragm, heart (further divided into atria and ventricles), lungs, BAT, and skin (cleared of hair using Topical Nair lotion, cleaned, then collected). At the same time, dissector three was collecting the pancreas, mesWAT, small intestine (further split into duodenum, jejunum, and ileum), cecum, and large intestine (further split into proximal and distal colon). As soon as tissue collection in the head was finished, the mouse carcass was cut transversely at the lumbar spine and handed to dissector one for muscle dissection. Dissector one then rapidly and precisely anatomized (bilaterally) the quadriceps (entire complex - rectus femoris, vastus lateralis, vastus intermedius, and vastus medialis), hamstrings (biceps femoris), gastrocnemius, plantaris, and soleus muscles. All tissues were washed with sterile 1× PBS to remove residual blood prior to snap-freezing. Additionally, the stomach, small intestine, and large intestine were cleared and cleaned of debris with PBS prior to freezing. The cecum, however, was kept whole containing all fecal/food matter and microorganisms present. All dissected tissues were snap-frozen in liquid nitrogen, and stored at –80°C for later analysis.

## Respirometry of frozen tissue samples

Respirometry was conducted on frozen tissue samples to assay for mitochondrial activity as described previously (*Acin-Perez et al., 2020*), using a Seahorse XFe96 Analyzer. Samples were thawed in 1× MAS buffer (70 mM sucrose, 220 mM mannitol, 5 mM KH$_2$PO$_4$, 5 mM MgCl$_2$, 1 mM EGTA, 2 mM HEPES pH 7.4), finely minced with scissors, then homogenized with a glass Dounce homogenizer on ice. The entire sample, bilateral for skeletal muscles, was homogenized and used for respirometry with the exception of liver in which only a piece was homogenized. This was done to provide a true tissue average and avoid regional differences in mitochondria. The resulting homogenate was spun at 1000×*g* for 10 min at 4°C. The supernatant was collected and immediately used for protein quantification by BCA assay (Thermo Scientific, 23225). Each well of the Seahorse microplate was loaded with the designated amount (in μg) of homogenate protein (*Table 1*). Each biological replicate was comprised of three technical replicates. Samples from all tissues were treated separately with NADH (1 mM) as a complex I substrate or succinate (a CII substrate, 5 mM) in the presence of rotenone (a CI inhibitor, 2 μM), then with the inhibitors rotenone (2 μM) and antimycin A (4 μM), followed by TMPD (also known as Wurster's reagent, 0.45 mM) and ascorbate (vitamin C, 1 mM) to activate CIV, and finally treated with azide (40 mM) to assess non-mitochondrial respiration.

## Quantification of mitochondrial content

Mitochondrial content of homogenates used for respirometry was quantified with the membrane potential-independent mitochondrial dye MitoTracker Deep Red FM (MTDR, Invitrogen, M22426) as described previously (*Acin-Perez et al., 2020*). Briefly, lysates were incubated with MTDR (1 μM) for 10 min at 37°C, then centrifuged at 2000×*g* for 5 min at 4°C. The supernatant was carefully removed and replaced with 1× MAS solution and fluorescence was read with excitation and emission wavelengths of 625 nm and 670 nm, respectively. All measurements were read with a BioTek Synergy HTX Multimode Plate Reader (Agilent). All samples from a single tissue-type were measured on the same plate, in duplicate, with an equal gain setting of 90 to ensure comparability across samples. To minimize non-specific background signal contribution, control wells were loaded with MTDR+1× MAS and subtracted from all sample values.

## Acknowledgements

This work was supported by the National Institutes of Health (DK084171 to GWW). DCS was supported by an NIH T32 training grant (HL007534).

## Additional information

### Funding

| Funder | Grant reference number | Author |
|---|---|---|
| National Institute of Diabetes and Digestive and Kidney Diseases | DK084171 | G William Wong |
| National Heart, Lung, and Blood Institute | HL007534 | Dylan C Sarver |

The funders had no role in study design, data collection and interpretation, or the decision to submit the work for publication.

### Author contributions

Dylan C Sarver, Conceptualization, Data curation, Formal analysis, Investigation, Visualization, Writing – original draft, Writing - review and editing; Muzna Saqib, Fangluo Chen, Investigation, Writing - review and editing; G William Wong, Conceptualization, Supervision, Funding acquisition, Investigation, Writing – original draft, Project administration

### Author ORCIDs

G William Wong ⓘ https://orcid.org/0000-0002-5286-6506

### Ethics

This study was performed in strict accordance with the recommendations in the Guide for the Care and Use of Laboratory Animals of the National Institutes of Health. All of the animals were handled according to approved institutional animal care and use committee (IACUC) protocols (# MO22M367) of the Johns Hopkins University School of Medicine.

Reviewer #1 (Public review): https://doi.org/10.7554/eLife.96926.4.sa1
Reviewer #2 (Public review): https://doi.org/10.7554/eLife.96926.4.sa2
Reviewer #3 (Public review): https://doi.org/10.7554/eLife.96926.4.sa3
Author response https://doi.org/10.7554/eLife.96926.4.sa4

## Additional files

### Supplementary files

• MDAR checklist

### Data availability

All data generated or analyzed during this study are included in the manuscript and supporting files; source data files have been provided for *Figure 1*.

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
