## [Editor Report · eLife Assessment]

This **important** study provides a comprehensive assessment of mitochondrial function across age and sex in mice. The strength of evidence supporting this resource is **compelling**, given the exhaustive number of tissues profiled and in-depth analyses performed.

---

## [Referee Report · Reviewer #1 (Public review)]

In this study, Sarver and colleagues carried out an exhaustive analysis of the functioning of various components (Complex I/II/IV) of the mitochondrial electron transport chain (ETC) using a real-time cell metabolic analysis technique (commonly referred as Seahorse oxygen consumption rate (OCR) assay). The authors aimed to generate an atlas of ETC function in about 3 dozen tissue types isolated from all major mammalian organ systems. They used a recently published improvised method by which ETC function can be quantified in freshly frozen tissues. This method enabled them to collect data from almost all organ systems from the same mouse and use many biological replicates (10 mice/experiment) required for an unbiased and statistically robust analysis. Moreover, they studied the influence of sex (male and female) and aging (young adult and old age) on ETC function in these organ systems. The main findings of this study are (1) cells in the heart and kidneys have very active ETC complexes compared to other organ systems, (2) the sex of the mice has little influence on the ETC function, and (3) aging undermined the mitochondrial function in most tissue, but surprisingly in some tissue aging promoted the activity of ETC complexes (e.g., Quadriceps, plantaris muscle, and Diaphragm).

Comments on the second revision:

My previous concern remains unaddressed in the new revision. As I mentioned earlier, it is crucial for the authors to include a detailed discussion on the limitations of their method, specifically how maximal respiration does not accurately reflect the actual ATP production rate. Additionally, the authors should highlight the fact that data provided in the manuscript should be interpreted with caution.

---

## [Referee Report · Reviewer #2 (Public review)]

Summary:

The authors utilize a new technique to measure mitochondrial respiration from frozen tissue extracts, which goes around the historical problem of purifying mitochondria prior to analysis, a process that requires a fair amount of time and cannot be easily scaled up.

Strengths:

A comprehensive analysis of mitochondrial respiration across tissues, sexes, and two different ages provides foundational knowledge needed in the field.

Weaknesses:

While many of the findings are mostly descriptive, this paper provides a large amount of data for the community and can be used as a reference for further studies. As the authors suggest, this is a new atlas of mitochondrial function in mouse. The inclusion of a middle aged time point and a slightly older young point (3-6 months) would be beneficial to the study.

---

## [Referee Report · Reviewer #3 (Public review)]

The aim of the study was to map, (a) whether different tissues exhibit different metabolic profiles (this is known already), what differences are found between female and male mice and how the profiles changes with age. In particular, the study recorded the activity of respirasomes, i.e. the concerted activity of mitochondrial respiratory complex chains consisting of CI+CIII2+CIV, CII+CIII2+CIV or CIV alone.

The strength is certainly the atlas of oxidative metabolism in the whole mouse body, the inclusion of the two different sexes and the comparison between young and old mice. The measurement was performed on frozen tissue, which is possible as already shown (Acin-Perez et al, EMBO J, 2020).

Weakness: The assay reveals the maximum capacity of enzyme activity, which is an artificial situation and may differ from in vivo respiration, as the authors themselves discuss. The material used was a very crude preparation of cells containing mitochondria and other cytosolic compounds and organelles. Thus, the conditions are not well defined and the respiratory chain activity was certainly uncoupled from ATP synthesis. Preparation of more pure mitochondria and testing for coupling would allow evaluation of additional parameters: P/O ratios, feedback mechanism, basal respiration, and ATP-coupled respiration, which reflect in vivo conditions much better. The discussion is rather descriptive and cautious and could lead to some speculations about what could cause the differences in respiration and also what consequences these could have, or what certain changes imply.

Nevertheless, this study is an important step towards this kind of analysis.

Comments on the second revision:

I believe this is an important and interesting area of study, although I recognise that the assay which measures maximal enzyme activity under unphysiological conditions has its limitations. Nevertheless, it does seem possible to get a first glance of the respiratory situation in the respective tissue. There is a typo in the source data (Fig. xC) for skeletal muscle.

---

## [Author Response]

The following is the authors’ response to the previous reviews.

**Recommendations for the authors:**

**Reviewer #2:**
No further questions, but please do add a sentence or two about the lack of these additional points in the discussion as a limitation to the study.

We have included additional “limitations of the study” in the Discussion Section.

**Reviewer #3:**
The authors have added to the discussion some critical remarks about the limitations of the study, which will help in the assessment of the conclusions.In sum, the manuscript has significantly improved during the revision.Some minor points should be changed, thoughPage 18 marked: "What causes an age-dependent decrease in mitochondrial OXPHOS genes across tissues, however, is largely unknown." I assume, the authors do not suggest that the abundance of genes is reduced, which means elimination of DNA? Be more precise about this.

We thank the reviewer for pointing this out. We have clarified this to mean “OXPHOS gene expression” and made a couple changes accordingly.

Page 18 marked : a paragraph was added addressing the increase in mitochondrial respiration in the heart, this should be discussed in the light of literature as it was done for skeleton muscle the following paragraph

We have included additional paragraphs in the Discussion Section to talk about increased mitochondrial respiration in the aging heart in the context of published literature.

Figure 2: it was asked for error bars for the OCR measurements. Response: We have added the error bars and statistical significance to revised Figure 2; however, is it correct that there are no significant differences?

Figure 2 ranks tissues based on the OCR values within a single group of mice (male or female, young or old) and is not a comparison between male vs female, or young vs old. For this reason, no statistics were included as they are not needed here. The goal of this figure is to highlight the OCR distribution across tissues within a single sex and age group.